# The Probability of Ship Collision during the Fully Submerged Towing Process of Floating Offshore Wind Turbines

**Yihong Li** , **Longxiang Liu, Sunwei Li** and **Zhen-Zhong Hu** *

Institute for Ocean Engineering, Shenzhen International Graduate School, Tsinghua University, Shenzhen 518055, China; y.li129@sz.tsinghua.edu.cn (Y.L.); liulx22@mails.tsinghua.edu.cn (L.L.); li.sunwei@sz.tsinghua.edu.cn (S.L.)
* Correspondence: huzhenzhong@tsinghua.edu.cn

**Abstract:** As global warming intensifies, the development of offshore wind farms is swiftly progressing, especially deep-water Floating Offshore Wind Turbines (FOWTs) capable of energy capture in deep-sea regions, which have emerged as a focal point of both academic and industrial interest. Although numerous researchers have conducted comprehensive and multifaceted studies on various components of wind turbines, less attention has been paid to the operational stage responses of FOWTs to wind, waves, and currents and the reliability of their structural components. This study primarily employs a theoretical analysis to establish mathematical models under a series of reasonable assumptions, examining the possibility of collisions between FOWT transport fleets and other vessels in the passage area during the towing process. Using the model, this paper takes the Wanning Floating Offshore Wind Farm (FOWF) project, which is scheduled to be deployed in the South China Sea, as its research object and calculates the probability of collisions between FOWTs and other vessels in three months from the pier near Wanning, Hainan, to a predetermined position 22 km away. The findings of the analysis indicate that the mathematical model developed in this study integrates the quantities and velocities of navigational vessels within the target maritime area as well as the speeds, routes, and schedules of the FOWT transport fleet. By employing statistical techniques and geometric calculations, the model can determine the frequency of collisions between various types of vessels and the FOWT transport fleet during the transportation period. This has substantial relevance for future risk assessments and disaster prevention and mitigation measures in the context of FOWT transportation.

**Keywords:** FOWTs; ship collision; geometric probability

## 1. Introduction

As energy consumption continues to rise, the depletion of conventional non-renewable energy resources is increasing [1]. The global climate crisis is accelerating, posing threats to human health and safety [2]. The overuse of fossil fuels is a major contributor to the crisis [3]. The efficient utilization of renewable clean energy and reductions in greenhouse gas emissions are the keys to achieving low-carbon sustainable development and addressing global challenges [4]. Wind power technology, with its minimal space requirements, low costs, high energy capture efficiency, and mature technical development, has become a global focus for new energy technology development [5]. However, due to limitations in land area and the completion of nearshore development, focusing on deepwater Floating Offshore Wind Turbines (FOWTs) has become the consensus in the industrial and scientific research communities [6]. Major industrialized countries around the world are actively developing offshore wind power, with the global installed capacity of offshore wind power setting new records repeatedly [7].

Thus far, extensive research on FOWTs has been conducted by numerous scholars, yielding distinguished outcomes. As Table 1 shows, these studies not only encompass the responses of wind, wave, and current loads on the mooring systems, floating foundations,

towers, and blades of FOWTs of various sizes but also include reliability analyses of the FOWTs' internal gearboxes and electrical systems, providing ample research materials and data for industrial practice. Furthermore, these studies predominantly focus on the operational phase of FOWTs, with only a few scholars [8,9] investigating the external risks encountered during the transportation of FOWTs to the designated site.

**Table 1.** FOWT-related research and content.

| Contents | Reference |
| --- | --- |
| Mooring system of FOWT | [10–12] |
| Foundation of FOWT | [13,14] |
| Structural dynamic response of FOWT | [15,16] |
| Blade system of FOWT | [17,18] |
| Reliability of FOWT | [19,20] |

However, relevant studies on the subject do exist and can be referenced. To investigate the potential risks associated with the transportation process of FOWTs, it is essential to have a comprehensive understanding of the types of FOWTs and the transportation processes involved. Firstly, with regard to the types of FOWTs, according to articles by Guedes Soares [21], Anders [22], and others, FOWTs are primarily categorized into three major types: the Spar type, Semi-Submersible type, and Tension Leg Platform (TLP), as illustrated in Figure 1. Each of these three types of FOWTs possesses distinct characteristics, leading to variations in the methods employed for their transportation to predetermined sites. Spar-type FOWTs typically employ a modular transportation approach in which the floating structure is first towed to the designated site and positioned, followed by the assembly of the blades, generator units, and towers at the dock. These components are then transported to the site and assembled through a docking installation, shown as Figure 2a. Semi-Submersible-type FOWTs are usually directly towed to the predetermined site by tugboats and positioned upon arrival, shown as Figure 2b. TLP-type FOWTs can be transported either on a barge or suspended beneath a vessel; they are transported to the designated site and subsequently positioned and installed, shown as Figure 3. Depending on the chosen engineering and transportation methods, the types of vessels used may include, but are not limited to, tugboats, crane barges, heavy lift cargo vessels, jack-up barges, purpose-built jack-up vessels, and semi-submersible crane vessels. Notably, methods such as the twin-hull floating and lifting techniques, which are still under validation, are not included in the aforementioned vessel types [23,24]. These differences contribute to substantial variations in the risks associated with the transportation process of FOWTs.

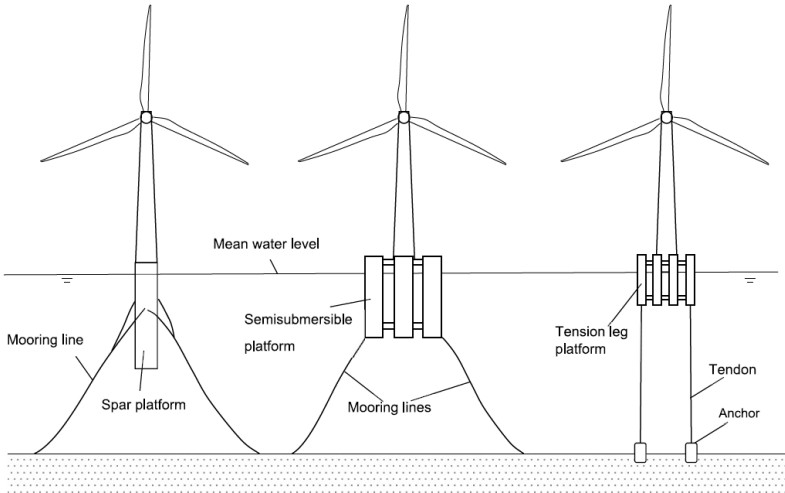

**Figure 1.** Three FOWT types [7].

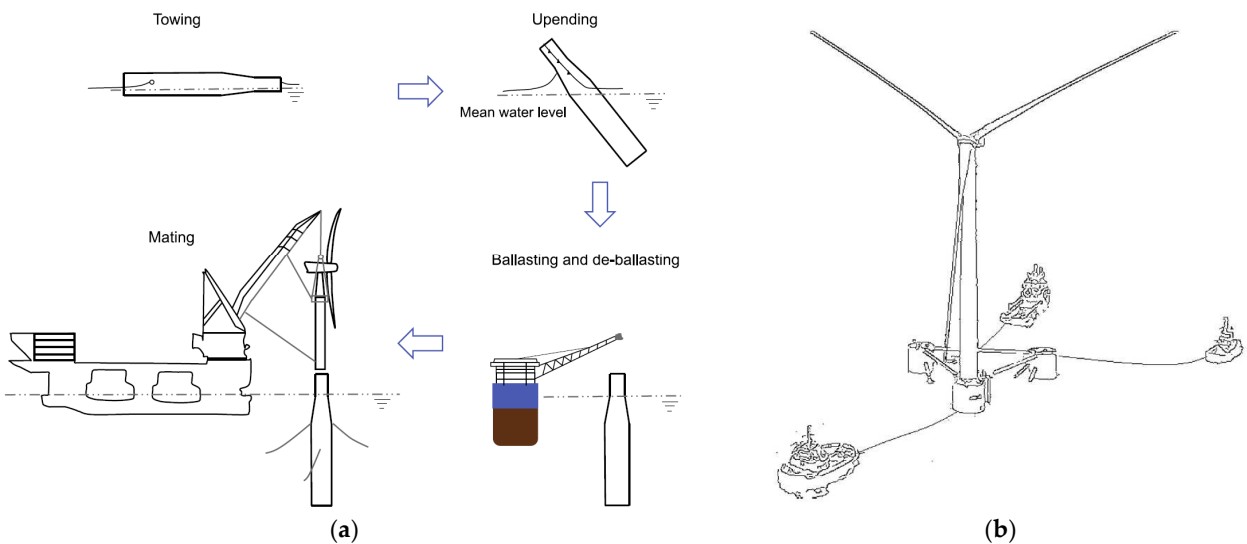

**Figure 2.** Transport methods of (**a**) Spar-type FOWTs; (**b**) Semi-Submersible-type FOWTs [7].

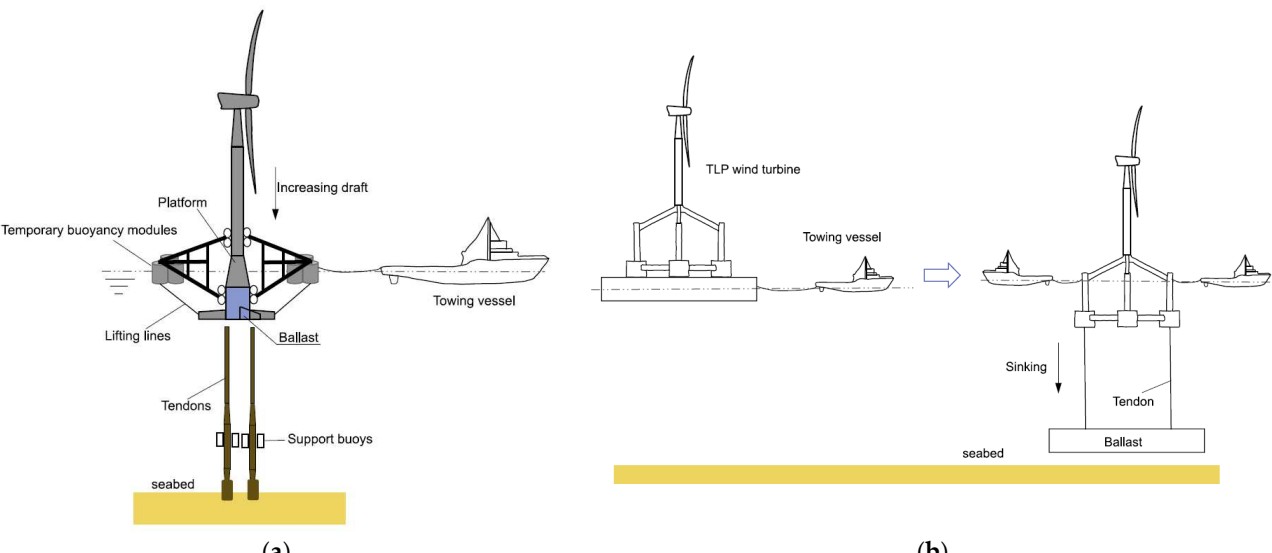

**Figure 3.** Installation methods of TLP FOWTs. (**a**) MIT type; (**b**) GICON type [7].

According to Xue [8] and Zhang's [9] papers, these differences are primarily manifested through various aspects, as follows:

- The towing process establishes a flexible connection between FOWTs and tugboats, implying that the relevant operations of the towing vessel cannot rigidly transmit to the FOWTs. Consequently, the motion of FOWTs is uncontrolled and primarily governed by inertia and resistance. In contrast, the transportation form provided by engineering vessels does not encounter such concerns as all components are on the engineering vessel and can be considered integral to the hull.
- Spar-type floating structures exhibit less conspicuous visual targets during the towing transport process, thereby increasing the likelihood of collisions with other vessels.
- Due to the higher center of gravity during the transportation procedure, Semi-Submersible and TLP-type FOWTs experience larger wave loads during the towing process, and these effects are more pronounced. This implies that these two types of FOWTs are less suitable for transportation via towing methods in regions with adverse sea conditions.

After gaining an understanding of the transportation processes associated with FOWTs, the exploration of relevant and analogous studies by other scholars becomes highly valuable. Presencia and Shafiee [25] conducted an analysis of engineering vessels during

construction and their collisions with wind turbines, summarizing numerous historical studies in the process. Their work reveals a shift in the research focus toward maritime vessel and structure collisions with wind turbines since 2005, with increasing diversity in the vessel types and collision scenarios considered, accompanied by the development of increasingly intricate mathematical models. Commonly employed approaches involve statistical distribution and frequency calculations to determine collision probabilities. LS-DYNA or another finite element simulation software is often utilized to simulate accident consequences. Furthermore, they provided a methodology for calculating the probability of collisions between engineering vessels and fixed offshore wind turbines, combining probability distributions with qualitative analyses to estimate the likelihood of different wind turbine components being struck by vessels.

Dai et al. [26] delved into the collision analysis of wind farm service vessels and fixed offshore wind turbines. Their research employed a standard Quantitative Risk Assessment (QRA) process, categorizing vessel types and collision modes and constructing event trees and accident trees and adapting them to Bayesian network algorithms utilizing directed acyclic networks. The consequence aspect of their study utilized a numerical simulation to obtain local and global yield limit energy values for various collision scenarios. Additionally, comprehensive research on ship-to-ship collisions has been well established, as demonstrated by Zhang's [27] thorough analysis and summary in his paper, encompassing both analytical methods and numerical simulations. The findings of these scholars provide a foundational basis and a conceptual framework for the present study.

This study focuses on employing an analytical approach to investigate the probability and consequences of vessel collisions during the transportation process of FOWTs, using the planned construction of a floating wind farm in the Wanning Sea area of China as a research subject. Mathematical models are developed to encompass various collision scenarios, collision intensities, and the post-collision motion of objects. This study's findings can assist engineers in selecting more suitable transportation methods based on the size and type of wind turbine, thereby enhancing construction safety.

## 2. Materials and Methods

### 2.1. Case and Site Introduction

This study selects an area under development for an FOWF in the eastern waters of Wanning City, Hainan Province, China, as the research site. This study focuses on 16~18 megawatt FOWTs deployed at this location, and all pertinent data can be found in Table 2 and Figure 4.

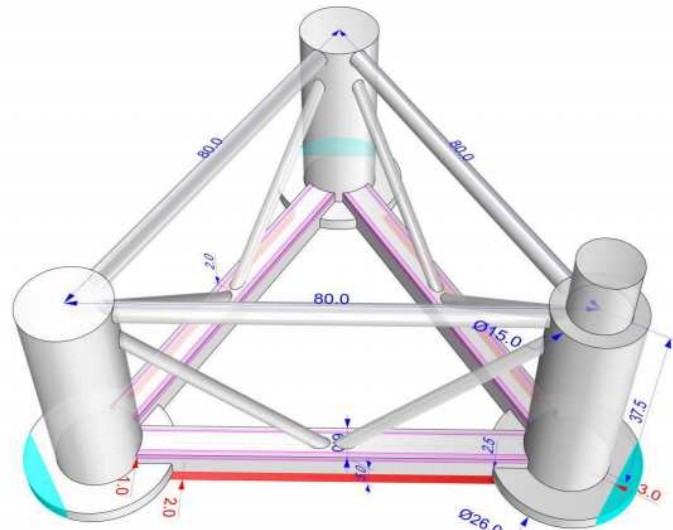

**Figure 4.** FOWT floater.

**Table 2.** Wanning FOWF project and FOWT information.

| Subjects | Value |
|---|---|
| Distance to port (km) | 22 |
| Water depth (m) | 100 |
| Ultimate angle of generator (°) | ±15 |
| Ultimate heave (m) | ±5 |
| Column radius (m) | 7.5 |
| Column height (m) | 37.5 |
| Swing plate radius (m) | 13 |
| Swing board height (m) | 2.5 |
| Column center distance (m) | 80 |
| Draft (m) | 25 |
| Blade size (m) | $123 \times 6.4 \times 5.4$ |
| Blade weight (t) | 55 |
| Nacelle size (m) | $16 \times 9.1 \times 14.1$ |
| Nacelle weight (t) | 416 |
| Hub size (m) | $8.8 \times 8.0 \times 7.8$ |
| Hub weight (t) | 124 |
| Tower height (m) | 126.2 |
| Tower weight (t) | 1132 |
| FOWT CoG (m) | 19.29 |
| FOWT CoB (m) | 8.89 |
| FOWT metacentric height (m) | 34.03 |

According to a report from the contractors of the FOWF, the project is located in coastal waters approximately 22 km east of Wanning City, Hainan Province, China. The average water depth in the area is approximately 100 m, and the site covers an area of approximately 160 km$^2$, as shown in Figure 5. The project is planned to be implemented in three stages, with a total of 66 FOWTs to be installed. The first stage is scheduled to commence commercial power generation in 2024, and the overall project will be completed and connected to the grid by the end of 2027. Due to the absence of freight ports in the vicinity of the FOWF region, as indicated by statistical data from the China Maritime Safety Administration's Automatic Identification System (AIS) [28], and in accordance with research findings by Wang [29,30], Wan [31], and Guan et al. [32], it has been observed that within a 50 km radius around the FOWF, there is a lack of major shipping channels. The predominant vessels navigating in this proximity are several cargo vessels, and most of them are small and mid-sized fishing boats, as evidenced by the results presented in Table 3.

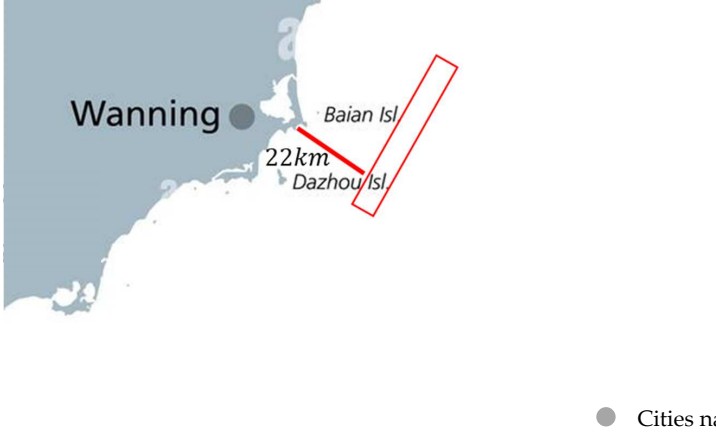

**Figure 5.** FOWF project area.

**Table 3.** Shipping volume and size near FOWF area [28–32].

| Subjects | Value |
|---|---|
| Lane number | 0 |
| Vessel number (/y) | 8766 |
| Vessel length (m) | 36.8 |
| Vessel breadth (m) | 6.8 |
| Vessel tonnage (t) | 229 |
| Vessel speed (m/s) | 5.2 |

Unfortunately, due to confidentiality concerns and the progression status of the project, a substantial number of detailed parameters are currently unavailable. Therefore, some of the data utilized in this study are derived from research conducted by other scholars in similar maritime areas or are sourced from data publicly disclosed on government websites.

*2.2. Ship Collision Analytical Model*

Similar to other scholarly inquiries into marine collision incidents, the probability computation model constructed in this paper is founded upon a set of assumptions pertaining to the transportation process of FOWTs. These assumptions are integrated with the regulations of the International Maritime Organization (IMO), the China Classification Society (China Class), and Rijkswaterstraat and are delineated as follows:

- Ship operators are expected to actively endeavor to prevent accidents.
- Ship operators are anticipated to adhere to applicable legal and regulatory frameworks.
- Vessels are assumed to commence their journey without any defective equipment.
- Transportation operations will not be conducted under adverse sea conditions or within the subsequent 24 h period.
- The likelihood of a collision resulting from objects falling from the intended target and floating on the sea's surface is not considered.

These assumptions effectively confine collisions to daylight hours advantageous for sea conditions and within a defined timeframe conducive to good visibility. This meticulously considered scope permits a concentrated analysis of collision risk assessment outcomes, emphasizing spontaneous and unforeseen causative factors.

To utilize a deductive method for constructing analytical models that estimate the collision frequency of FOWT transport fleets and other vessels, it is essential to understand the various scenarios in which collisions occur. In this study, the transport fleet is considered a flexible interconnected vessel, such that collisions between the fleet and other vessels are simplified into ship-on-ship collisions. Based on an investigation of the Pedersen model, in combination with engineering practice, three primary collision scenarios during transportation are identified:

- Collisions occur as the fleet traverses the shipping channel.
- Collisions happen as the fleet approaches the shipping channel.
- Collisions take place in non-channel areas of the sea.

Additionally, an analysis of the mainstream Pedersen method reveals that models for calculating the frequency of vessel collisions are mainly divided into two parts: the collision candidate segment and the fundamental probability segment. Therefore, the frequency estimation of collisions between FOWTs and other vessels during towing transportation should conform to the analysis and calculation process presented in Figure 6.

Since it is challenging to estimate the posture of vessels just prior to and during a collision, this study assumes that a collision occurs once the geometric circles representing the collision zones of two ships touch. Equations (1) and (2) provide a methodology for calculating the diameter of the collision area for ships and FOWTs.

$$W_S = 2/\pi(L_S + B_S) \tag{1}$$

$$W_T = 4\sqrt{3}/3\left(\frac{D_f}{2} + R_f\right) \tag{2}$$

where $W_S$ is the equivalent width of the ship; $L_S$ is the length of the ship; $B_S$ is the breadth of the ship; $W_T$ is the equivalent width of the FOWT; $D_f$ is the distance between the center of the floaters; and $R_f$ is the radius of floaters.

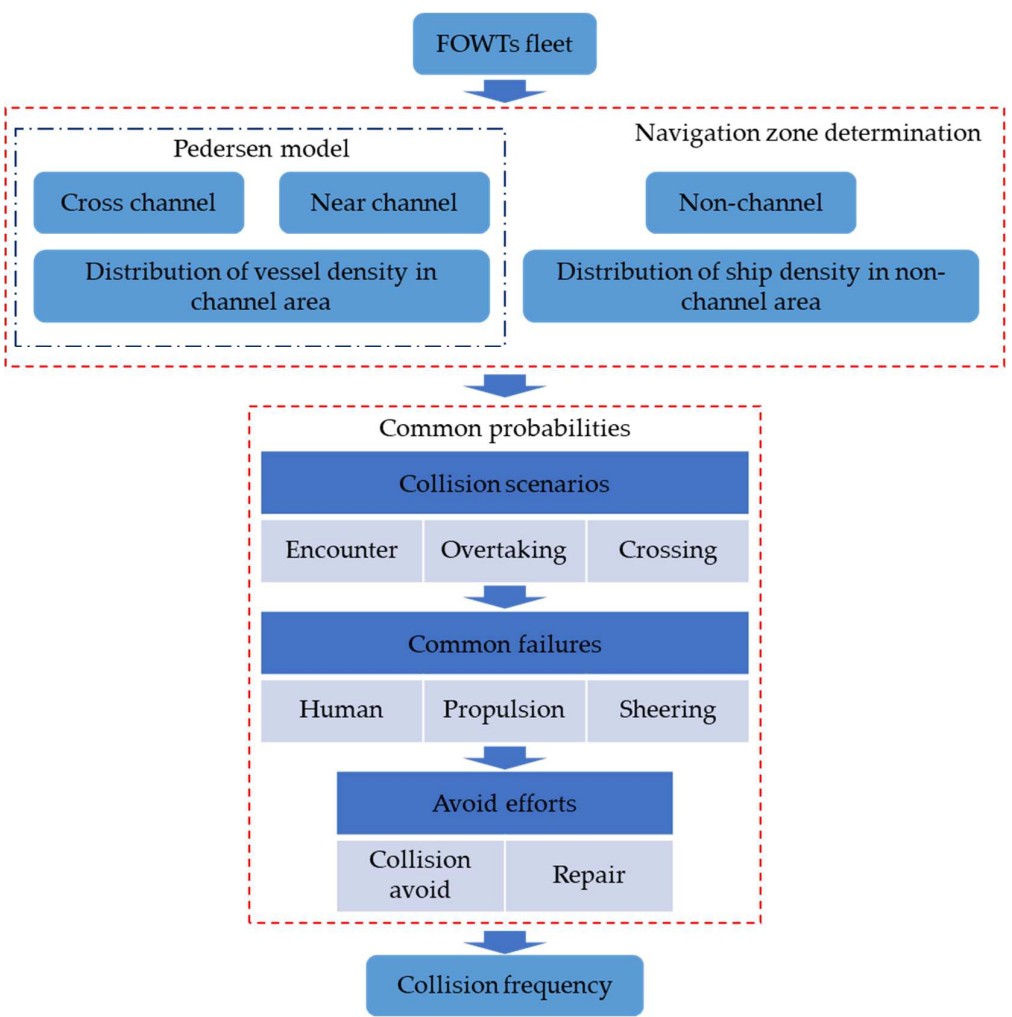

**Figure 6.** Flowchart of FOWT transportation fleet collision frequency estimation.

Lastly, consideration is given to the methodology employed for probability calculations. To analyze the reasons behind collisions between two vessels, it is essential to investigate the fundamental causes of collision incidents. Table 4 presents a series of fundamental reasons for vessel collisions, along with the corresponding symbols used in this study, the respective values employed, and the origins of these numerical values.

$$P_r(t_e) = 0.5/0.605\left(t_e/0.605\right)^{0.4} e^{-\left(t_e/0.605\right)^{0.5}} \tag{3}$$

$$t_e = D_E/V_{ij} \tag{4}$$

where $V_{ij}$ is the relative speed of ships $i$ and $j$; $t_e$ is the evasive duration; and $D_E$ is the evasive distance, set as 2.5 km. The Equations (3) and (4) are used to calculate the duration it may take for ships to make evasive reactions when they see each other. If there is no evasive reaction on either side, a collision accident is highly likely to happen.

**Table 4.** Failure rates of ship devices and human failure rate.

| Subjects | Symbol | Value | Source |
|---|---|---|---|
| Human failure | $P_{Human}$ | $2 \cdot 10^{-4}/h$ | [33,34] |
| Propulsion failure | $P_{pf}$ | $1.5 \cdot 10^{-4}/h$ | [35] |
| Sheering failure | $P_{sf}$ | $6.3 \cdot 10^{-5}/h$ | [35] |
| Collision avoids | $P_{ca}$ | 0.5 | [36] |
| Repaired distribution | $P_r(t_e)$ | Equation (3) | [35] |

2.2.1. Channel and Near-Channel Area Collision Analytical Model

As delineated earlier, if there is an overlap between the transportation route and the primary navigational channels, it necessitates consideration of potential collisions occurring separately for vessels in the navigational channels and the transportation fleet. In such instances, adhering to the Pedersen method, it is posited that the navigational behavior of vessels in the channel follows a normal distribution. Conversely, the towed transport fleet is conceptualized as an obstacle possessing a certain level of active avoidance capability—a circumstance not addressed in the Pedersen method, which assumes static obstacles without evasion functionalities along the route. According to the description in Figure 7, this situation gives rise to four major collision categories: encounter, overtaking, crossing, and bend collisions. The bend collision category can be further subdivided into opposite-direction and same-direction scenarios, show as Figure 8 [27]. Consequently, the ensuing deductive analysis commences by examining and modeling these scenarios individually.

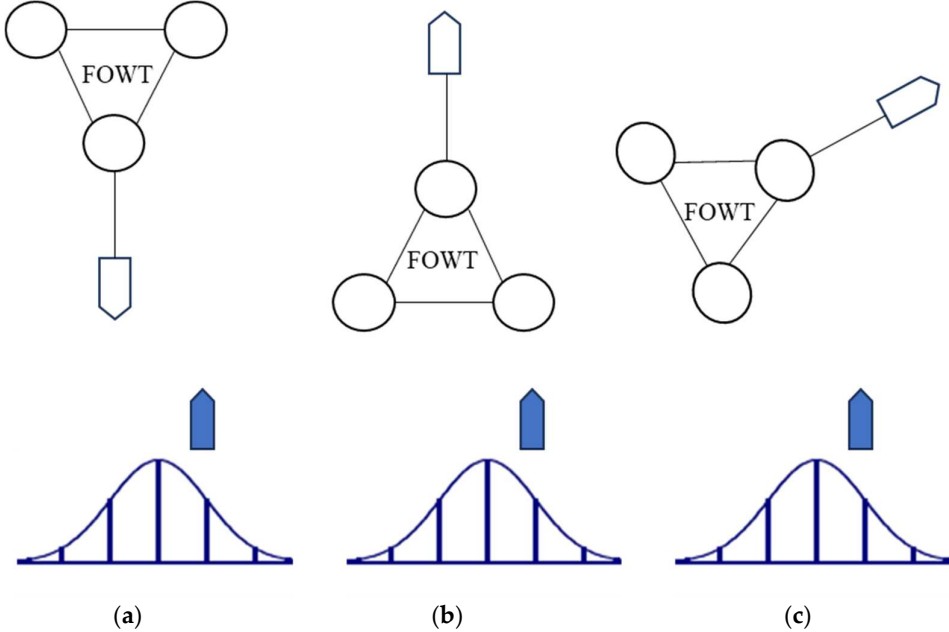

(**a**)            (**b**)            (**c**)

**Figure 7.** Three collision scenarios: (**a**) encounter collision; (**b**) overtaking collision; (**c**) crossing collision [27].

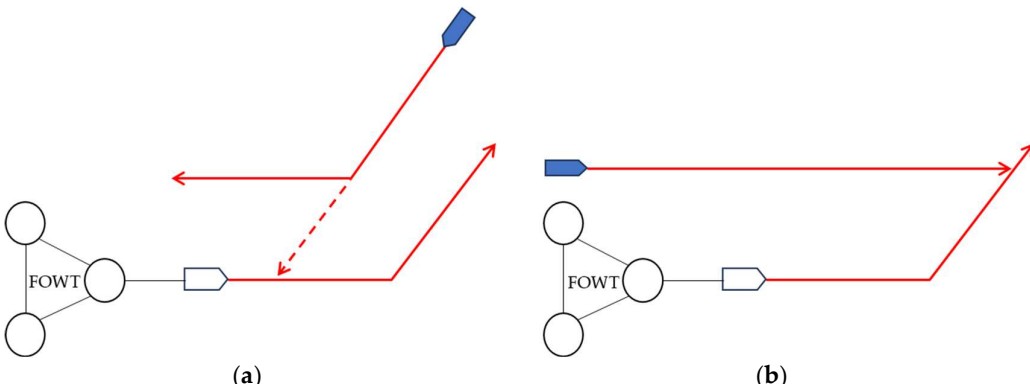

**Figure 8.** Two bend collision types: (**a**) opposite direction; (**b**) same direction [27].

Firstly, analyze the three scenarios depicted in Figure 7. For vessels navigating in the channel, their navigational behavior typically conforms to a normal distribution, as illustrated in Equation (5), with Figure 9 providing supplementary clarification of this normal distribution. In the event of contact between two vessels or a vessel and an FOWT, it can be assumed that the collision happened when the geometric circles formed by their main scales are tangent. The length of this segment is defined as the geometric collision diameter. Formula (6) represents the geometric diameter for vessel-to-vessel collisions, and Formula (7) represents the geometric diameter for vessel-to-FOWT collisions. Thus, the time elapsed before an incident occurs, after a vessel in channel *i* detects the towed transport fleet, is determined by the reaction distance between the two vessels, the relative velocity, and the geometric collision diameter. Expression (8) provides a function for the number of collisions between vessels on channel *i* and the fleet within a unit of time. By substituting Equations (5)–(7) into (8), the total number of vessels colliding with the fleet in channel i within time t can be obtained, as expressed in Formula (9). It is noteworthy that when investigating encounter and overtaking collisions, the sequence of vessel collisions needs to be considered. Therefore, appropriate modifications to the term $(D_{S-S} + D_{S-T})$ in the equation are required, reflecting that vessels in the channel cease to collide with the fleet after encountering the first object in the towed transport fleet.

$$f(z) = 1/\sqrt{2\pi}\sigma exp\frac{-(z-\mu)^2}{2\sigma^2} \tag{5}$$

$$D_{S-S} = 2/\pi(L_{S1+S2} + B_{S1+S2}) \tag{6}$$

$$D_{S-T} = 2/\pi(L_S + B_S) + 4\sqrt{3}/3\left(D_f/2 + R_f\right) \tag{7}$$

$$N_i = Q_i/v_i f_i(z_i)(D_{S-S} + D_{S-T})V_{ij}dz_i dt \tag{8}$$

$$N_t = \sum_i \int_0^t \int_{z_i} Q_i/V_i 1/\sqrt{2\pi}\sigma exp\frac{-(z_i-\mu)^2}{2\sigma^2}(D_{S-S} + D_{S-T})V_{ij}dz_i dt \tag{9}$$

where $z$ is the distance from the centerline of the navigation route; $\mu$ is the mean value of the spatial distribution and $\sigma$ is the standard deviation; $D_{S-S}$ is the geometrical collision diameter between ships; $D_{S-T}$ is the geometrical collision diameter between the ship and the FOWT; $L_{S1+S2}$ is the total overall length of ships; $B_{S1+S2}$ is the total breadth of ships; $N_i$ is the collision candidate in shipping route *i* during the unit period; $Q_i$ is the quantity of ships in the shipping lane; $V_i$ represents the velocities of the ships in the shipping lane; $N_t$ is the total number of collision candidates in the shipping lane as the transportation fleet passes the shipping lane; and *t* is the duration of the transportation fleet passing the shipping lane.

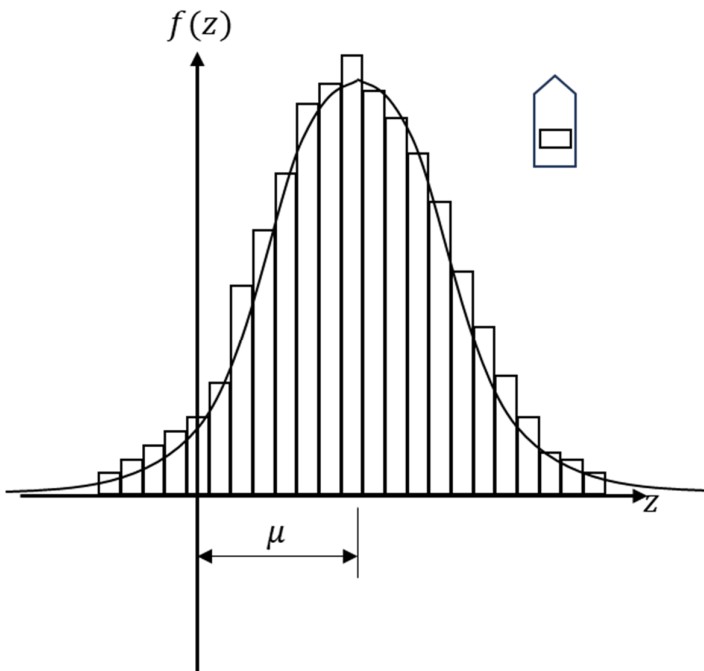

**Figure 9.** Lateral distribution of ship traffic.

Once the total number of vessels potentially involved in collisions within the route is determined, the collision frequency can be calculated by combining the probabilities corresponding to various collision forms. Equation (10) represents the calculation method for the frequencies of three common collision forms, where the value of $P_c$ can be referenced from Table 5.

$$F_{collison} = N_t \cdot P_c \tag{10}$$

**Table 5.** Causation probabilities from observations [37].

| Subjects | $P_c$ |
|---|---|
| Encounter | $4.9 \times 10^{-5}$ |
| Overtaking | $1.1 \times 10^{-4}$ |
| Crossing | $1.29 \times 10^{-4}$ |

The intricacies of bend collision scenarios, as illustrated in Figure 8, introduce a level of complexity. Traditionally, the establishment of a reliable analytical model necessitates considerations such as the operational status and inspection frequency of the Vessel Traffic System (VTS), as well as factors including the reaction distance between vessels. The integration of these diverse factors makes it challenging to formulate a robust analytical model. Consequently, when scrutinizing this particular issue, it is plausible to posit that occurrences of such accidents are rooted in anomalies with vessels navigating the channel itself. Within the temporal window subsequent to detecting the towed transport fleet, these vessels fail to execute appropriate course adjustments, and the fleet is unable to evade successfully. This leads to the formulation of Equations (11) and (12). Additionally, given the inherent absence of evasion capabilities in FOWTs, the analysis of collisions between navigating vessels and FOWTs obviates the need to consider fleet evasion scenarios.

$$F_{collision} = N_t \cdot P_{Human} \cdot (1 - P_{ca}) \text{ for opposite direction bend collision} \tag{11}$$

$$F_{collision} = N_t \cdot P_{Human} \cdot (1 - P_{ca})^2 \text{ for same direction bend collision} \tag{12}$$

The aforementioned model employs a deductive approach to analyze and construct a frequency estimation model for collisions between FOWT transport fleets and vessels in navigational channels. The analytical process reveals a multitude of factors that must be considered when assessing vessel collisions. In practical engineering applications with similar requirements, it is advisable to deploy safety vessels on the periphery of the fleet to alert passing vessels to alter their course, thereby mitigating the risk of accidents.

2.2.2. Non-Channel-Area Collision Analytical Model

For towed projects not situated near navigational channels, the methods mentioned earlier cannot be applied to estimate collision frequencies. The navigational behavior of fishing vessels often involves repetitive movements within a specific region, while engineering vessels navigate linearly between operation sites and docks, avoiding impassable areas. Both behaviors are challenging to statistically analyze and predict due to their unique characteristics. Consequently, a reanalysis of collision scenarios between these two types of vessels is necessary, and the specific analytical approach is illustrated in Figure 10.

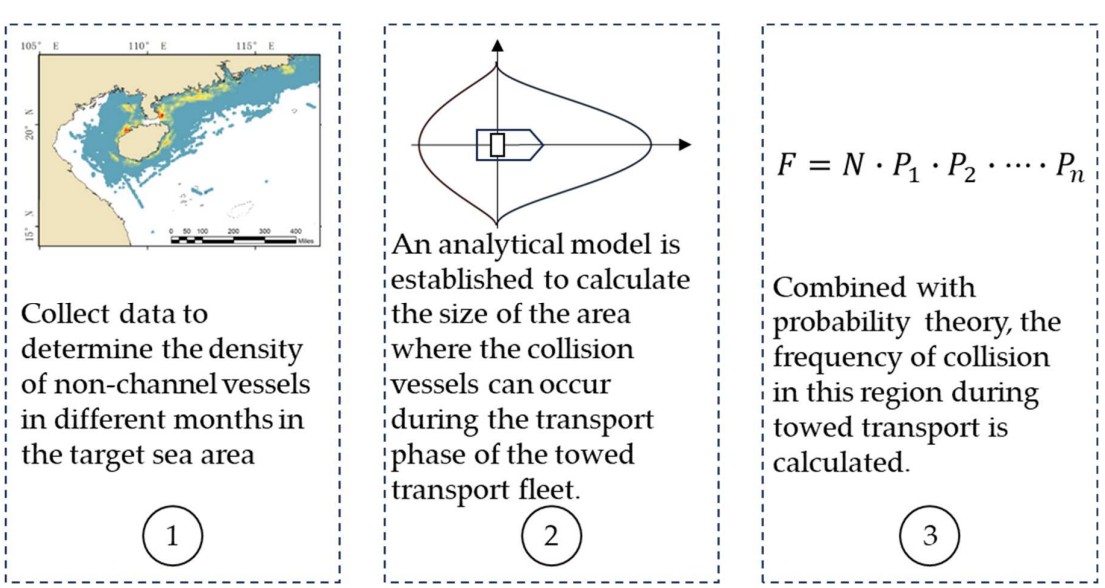

**Figure 10.** The steps of establishing an analytical model of ship collision in a non-channel area.

Firstly, an estimation of the total number of vessels that could potentially collide in a given maritime area needs to be conducted. This estimation should be based on monthly statistics, taking into account the differences in fish species present in the sea during different months, as well as restrictions imposed by fishing bans in the vessels' respective countries. These factors can lead to significant variations in the number of fishing vessels, vessel types, and concentrated distribution areas in a particular maritime region. In this study, an assumption of uniform vessel density in a given maritime area is employed.

Next, the calculation involves determining all vessels that could potentially navigate into the path of the transport fleet during transportation. Essentially, this entails a two-dimensional quantification of the overtaking problem: calculating the range within which all nearby vessels, traveling at speeds ranging from $-v_2$ to $v_2$ (where $v_2 > v_1$, the speed of the transport fleet), would converge towards the transport fleet from various directions during the transportation process at the speed of $v_1$.

Finally, when a navigating vessel visually detects or radar scans the towed transport fleet on its predetermined course, an avoidance response should be initiated. In the event of a collision, a malfunction in the propulsion and/or steering systems of the navigating vessel is indicated. Therefore, the probability of such occurrences should be factored into the analysis.

In summary, Equations (13) to (16) provide a mathematical analytical model for collisions of this nature.

$$\rho_{pmi} = \sum_m \sum_i N_{mi}/A_p \tag{13}$$

$$A_{candidates} = \int_{(v_2-v_1)t}^{(v_2+v_1)t} \int_{-v_2t}^{v_2t} \sqrt{x^2 + y^2 - 2xy\cos[\theta(t)]}dxdy \tag{14}$$

$$\theta(t) = \int_{0°}^{360°} \cos^{-1}v_1\left(1 - t^2\right)/v_2 \, dt \tag{15}$$

$$F_{collision} = \frac{1}{360} \cdot \rho_p \cdot A_{candidates} \cdot \left\{ P_c + P_{pf}[1 - P_r(t_e)] + P_{sf}[1 - P_r(t_e)] \right\} \tag{16}$$

where $\rho_{pmi}$ is the *i*-type ship density in *m* month in the project area; $N_{mi}$ is the ship number of *i*-type ships in *m* month; $A_p$ is the project area; $A_{candidates}$ is the ship location area, which may host a collision with the transport fleet when ships travel at velocity $v_2$; $v_1$ is the fleet velocity; $x$ is the horizontal displacement of ships; $y$ is the vertical displacement of ships; $t$ is the duration of the fleet from the port to the wind farm; and $F_{collision}$ is the frequency of collision. Figure 11 helps us understand the process of building the analytic model. Assuming that during the process of a fleet moving from point *A* to point *B* at a velocity $v_1$, there is a ship approaching the fleet at a velocity ranging from *0* to $v_2$ at each $\Delta t$ moment. Then, within the total duration *t* from point *A* to point *B*, the cumulative area of all the positions at which ships could potentially collide with the fleet is as depicted in the figure. In this process, the direction angle of the domain vessel $v_2$ varies with time, and the relative position between the two ships also diminishes as time elapses. The analysis process did not take into account the collision geometry between the ship's hull and the FOWT as the collision geometry of ships and FOWTs is negligible at the scale of the sea area, rendering it insignificant for computational purposes.

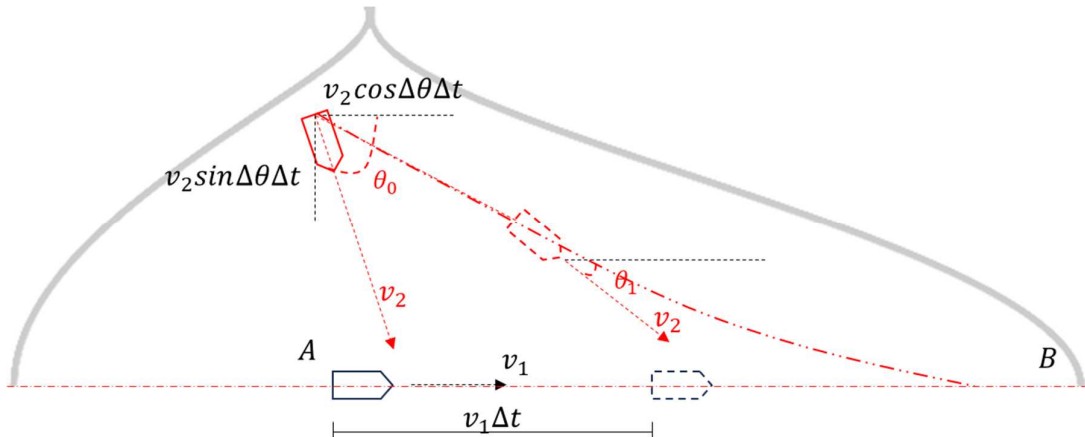

**Figure 11.** Establishment of the analytical model for the collision-detectable area.

According to the above analytic formula, the collision frequency between the transport fleet and ships in a non-channel area can be roughly calculated.

In summary, the analytical scale and the methods applied by the fleet in navigating through and avoiding channel routes exhibit significant differences. The fundamental causes of these differences lie in the distinct characteristics of the navigation channels and the traffic volume: the navigational behavior and traffic volume of vessels in primary channels are more predictable compared to those in fishing zones, with a considerable amount of research having utilized various methods to analyze them; hence, collision probability calculations can be conducted on a smaller scale using relatively mature distribution methods. In contrast, the behavior of vessels in non-channel areas, particularly fishing

boats, is more challenging to statistically analyze and predict, necessitating the estimation of collision probabilities on a larger scale through geometric analysis techniques.

## 3. Results

As previously mentioned at the onset of the second section, the Wanning FOWF project is situated in a vicinity devoid of any major shipping channels, with the site and its proximate areas primarily serving coastal fishing vessels. Consequently, in the case study analysis presented herein, collisions involving shipping fleets transiting or proximate to channels are not considered. The analysis is confined solely to scenarios of collisions between fishing vessels and shipping fleets. Due to the lack of publicly available monthly statistics on the number of fishing vessels in various sea areas from Chinese official sources, this paper utilizes data for the months of April, June, and September published by Guan et al. [32] in their research, with detailed data presented in Table 6. According to the data, the target sea area encompasses approximately 945,000 km$^2$. Considering that most vessels operate in nearshore regions, the actual fishing vessel distribution area is delimited to a 150 km offshore radius, resulting in a target sea area of approximately 350,000 km$^2$.

**Table 6.** Number of vessels in the project area [32].

| Months | Number of Vessels | | | | |
|---|---|---|---|---|---|
| | **Trawler** | **Gillnetter** | **Seiner** | **Other** | **Summary** |
| April | 2636 | 4085 | 1268 | 1317 | 9306 |
| June | 639 | 627 | 557 | 387 | 2210 |
| September | 1638 | 4562 | 1027 | 1805 | 9032 |

In this study, the speed of the towed transport fleet is set at 4 knots (7.4 km/h), and the cruising speed of fishing vessels is set at 9–12 knots (16–22 km/h). After the calculation, the number of collisions during transportation for each of three months is presented in Tables 7–9.

**Table 7.** Collision number during fleet transportation in April.

| Months | Collision Type $\times 10^{-3}$ | Ship Velocities (Knots) | | | |
|---|---|---|---|---|---|
| | | **9** | **10** | **11** | **12** |
| April | Encounter | 1.42 | 2.05 | 2.83 | 3.78 |
| | Overtaking | 1.86 | 2.68 | 3.70 | 4.94 |
| | Crossing | 2.00 | 2.88 | 3.97 | 5.30 |

**Table 8.** Collision number during fleet transportation in June.

| Months | Collision Type $\times 10^{-3}$ | Ship Velocities (Knots) | | | |
|---|---|---|---|---|---|
| | | **9** | **10** | **11** | **12** |
| June | Encounter | 0.34 | 0.49 | 0.67 | 0.90 |
| | Overtaking | 0.44 | 0.64 | 0.8 8 | 1.17 |
| | Crossing | 0.47 | 0.68 | 0.94 | 1.26 |

**Table 9.** Collision number during fleet transportation in September.

| Months | Collision Type $\times 10^{-3}$ | Ship Velocities (Knots) | | | |
|---|---|---|---|---|---|
| | | **9** | **10** | **11** | **12** |
| September | Encounter | 1.38 | 1.99 | 2.75 | 3.67 |
| | Overtaking | 1.80 | 2.60 | 3.59 | 4.79 |
| | Crossing | 1.94 | 2.79 | 3.85 | 5.14 |

The results indicate that the analytical model employed in this study effectively calculates the estimated number of collisions involving vessels of various speeds within

the target area approaching from multiple directions and encountering fleets during the towing process. This suggests that the equations constructed in this research study possess practical utility. Subsequent sections will provide a detailed discussion of the numerical contents of the outcomes.

## 4. Discussion and Limitation

As discussed in the literature review section of this paper, although there is considerable research on collisions between vessels and FOWTs during transportation, these studies commonly focus on the analysis of the FOWTs' dynamic behaviours and structural responses. The calculation of relevant collision probabilities has typically employed Pedersen's ship collision model. Given that the case study associated with this research is located in waters away from shipping lanes and that the Wanning FOWF project had not entered the construction phase by the date of this paper's completion, the validation component of this study treated the transportation fleet as a single vessel. It utilized research on ship collisions conducted by Srđan et.al. [38] in a similar maritime region as validation.

Srđan employed AIS data, incorporating a Monte Carlo simulation and Bi-LSTM (Bidirectional Long Short-Term Memory) neural networks to investigate the annual frequency of vessel collisions in the vicinity of Split. Their findings revealed that the average annual frequency of vessel collisions in heavily trafficked channels was approximately between $8 \times 10^{-3}$ and $1 \times 10^{-2}$, whereas in non-channel areas, the average frequency of vessel collisions was about $5 \times 10^{-3}$ or lower. Table 10 presents a comparison of the computational outcomes of the model developed in this study with the results obtained by Srđan and his colleagues.

**Table 10.** Comparison and validation.

| Results | Channel Area $\times 10^{-3}$ | Non-Channel Area $\times 10^{-3}$ |
|---|---|---|
| Srđan et.al. | $\approx$9.00 | <5.00 |
| This paper | - | 6.88 |

The comparison indicates that the analytical model constructed in this research yields collision frequency assessments for non-channel areas that are very close to those of Srđan, with numerical discrepancies likely arising from five factors:

- Variations in vessel density across different sea areas.
- Significant differences in the sizes of vessel collisions, with the present study's collision sizes considering scenarios in which transport vessels and FOWTs collide with other vessels.
- Methodologically induced numerical differentiation accounts for some discrepancies.
- Limitations in data usage for this study led to some bias in the results.

This comparison demonstrates the feasibility of the collision frequency calculation model for FOWTs towed by transport vessels and other vessels in non-channel areas, as presented in this study. However, situations involving transiting through channels or transportation activities in the vicinity of channels are not covered by this study; thus, actual data verification is not feasible. Nevertheless, the Pedersen method has been confirmed as feasible by multiple studies; when investigating collisions involving vessels and FOWT transport fleets, treating the transport fleet as a unified entity allows for the application of the Pedersen method.

In accordance with the FOWT transportation fleet collision probability assessment model for non-navigational areas proposed in this research, and in conjunction with the Pedersen probability assessment model, it is the authors' contention that the establishment of a mathematical model for a collision probability assessment necessitates particular focus on three primary aspects.

Firstly, the geometric and dynamic characteristics of the entities involved in potential collisions, which encompass vessels, platforms, floating debris, and reefs, among oth-

ers, should be considered. These parameters are directly related to the complexity of collision scenarios.

Secondly, attention must be directed toward the density of collision candidates within the study area. Presently, there are two ideal methods: the first involves the use of a distribution probability form, as employed in the Pedersen model; the second method entails the utilization of AIS data for statistical analysis.

Lastly, there is the estimation of the collision candidates' own navigation, piloting, and human error probabilities, which indeed represent the most stochastic component. Currently, the only reliable approach is to estimate these probabilities through the use of random algorithms or statistical outcomes.

However, these primary aspects also bring limitations to this kind of mathematical model:

- Currently, all collision scenarios are simplified in that they adequately reduce the geometric and kinematic characteristics of collision candidates—that is, they do not consider three-dimensional space and focus only on the two-dimensional plane; they do not account for changes in speed and assume a constant velocity; and they significantly simplify the detailed geometric dimensions of the collision candidates. This necessarily leads to biases in the assessment results. However, there are currently no studies that demonstrate the severity of the impact of these biases;
- AIS data that correspond to the geometric and dynamic characteristics of collision candidates is often difficult to obtain; thus, statistical research in this area is more appropriately conducted by shipping professionals or organizations;
- Due to the current limitations in scientific and technological advancements, the methods for estimating human error and equipment failure probabilities in existing collision probability assessment models can be considered rather rudimentary. The implementation of digital twin technology and artificial intelligence for personnel and equipment reliability monitoring is expected to significantly enhance the estimation of such data.

Upon verifying the viability of the model introduced in this paper, a comprehensive analysis of the model itself and the resulting data highlights five aspects that necessitate significant consideration:

- The longer the towing process, the higher the probability of ship collisions.
- The earlier ships in the domain detect the transport fleet, the more likely they are to avoid collision accidents.
- The slower the towing speed, the more likely ship collisions are.
- The faster the ships in the domain sail, the more likely they are to experience collisions.
- The ranking of the danger levels for collision types is as follows: crossing > overtaking > encounter.

Furthermore, through the analysis of the evaluation data, this paper provides some disaster prevention and mitigation suggestions corresponding to the case:

- Conduct transport operations during good visibility conditions as much as possible.
- The fleet should have conspicuous markers to warn nearby ships to give way.
- If possible, arrange for peripheral patrol ships for the fleet as a warning.
- The speed during the transportation process should not be too low.
- The transport route design should be as short as possible, and the channel should be avoided as much as possible.
- Collision buffer devices should be installed on the side of the ship and the FOWT to minimize the impact force of collisions.
- Conducting transport operations during the fishing off season is safer than transporting during the fishing season.

## 5. Conclusions

Generally speaking, this paper focuses on the Wanning FOWF project in Hainan and innovatively constructs an analytical model for ship collision probability in non-channel areas, building upon the Pedersen ship collision probability calculation model. The newly

developed model has been validated to estimate the frequency of collisions between vessels and FOWT-towing fleets in non-channel areas during the towing transportation of FOWTs. By analyzing the results of the model, this paper outlines some considerations for the towing transportation process of FOWTs, providing a model-based reference for further research into the risks associated with FOWT-towing transportation.

In the future, this study will advance in conjunction with the incremental progression of the Wanning FOWF project. The research will encompass, but will not be limited to, the following areas:

- The validation of the feasibility of a non-channel analysis model through the utilization of more extensive site data and actual statistical data.
- An enhancement of the existing Pedersen model and the collision probability calculation model for non-channel areas by considering factors such as ship draft and the height of FOWT structures, resulting in a three-dimensional ship collision probability calculation model.
- Further research into the distribution patterns of offshore fishing vessels will be conducted, utilizing mathematical methods for induction, to derive a distribution function. This distribution function, obtained through induction, will replace the average distribution assumption employed in this study.
- The optimization of the collision probability assessment algorithm through the integration of more advanced methods tailored to the practical circumstances of the shipping industry, aiming to prevent the occurrence of collision events.
- The formulation of targeted preventive and mitigation strategies for accidents during the towing transportation of FOWTs, based on the analysis results of the model.

**Author Contributions:** Conceptualization, methodology, data curation, and writing—original draft preparation, Y.L.; writing—review and editing, L.L.; supervision, S.L.; writing—review and editing and supervision, Z.-Z.H. All authors have read and agreed to the published version of the manuscript.

**Funding:** This research was funded by Guangdong Basic and Applied Basic Research Foundation, grant number 2022B1515130006; and Major Program of Stable Sponsorship for Higher Institutions (Shenzhen Science and Technology Commission, grant number WDZC20200819174646001.

**Data Availability Statement:** All data have been presented in the paper. Please refer to this article or citation.

**Acknowledgments:** We express our gratitude to the Power China Guizhou Engineering Co., Ltd. for the partial data support provided to this research study.

**Conflicts of Interest:** The authors declare no conflicts of interest.

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
