# Peer review of "The Probability of Ship Collision during the Fully Submerged Towing Process of Floating Offshore Wind Turbines"

_sustainability, doi:10.3390/su16041705_

Round 1
Reviewer 1 Report
Comments and Suggestions for Authors
The paper is well-written and well-organized. Some suggestions are as follows:
In abstract, "The findings of the analysis suggest that the mathematical model has the capability to efficiently determine the frequency of collisions between vessels within the domain and the transport team during the transportation process. " It should explain why and how to determin the frequency.
Floating Offshore Wind Farm (FOWF) appears many times in the abstract as well as in the text. Just use it in the first appearance and use abbreviation afterwards.
In table 1, switch the first and second column would be better.
Soares [20] shoudl be "Guedes Soares [20]"
Can you give some quantitative analysis on the discussion section?
Author Response
Please find the attached file, Thanks.

Reviewer 2 Report
Comments and Suggestions for Authors
The aim of this article was to conduct the assessment of probability of ship collision during the fully submerged towing process of floating offshore wind turbines from the total assembly pier in Hainan Wanning to the predetermined position 22 km away on Wanning Offshore Floating Wind Farm (FOWF), which is scheduled to be deployed in the South China Sea, as the research object. This study primarily employs theoretical analysis to establish mathematical models under a series of reasonable assumptions, examining the possibility of collisions between FOWT transport fleets and other vessels in the passage area during the towing process. This model repeatedly takes into account the waters through which the FOWT towing process may pass, possible collision scenarios, and takes into account the possibility of collisions between the ship and the FOWT transport fleet of different months and speeds. From the conducted research, the authors concluded that June may be the best moment for FOWT to join the Wanning FOWF project. In addition, appropriate practitioners must pay special attention to enhancing the target visibility of the fleet during the transportation process and avoid excessively slow transportation speeds. Summing up, the whole paper is clear, well-presented and fully understandable. Figures and Tables included in the paper are well prepared. There is also a direct relationship between the title and the paper. The entire text constitutes a well-written and very interesting contribution for MDPI. I have no substantive comments and I appreciate the study. I wish the authors good luck and further success in their research.
Reviewer 3 Report
Comments and Suggestions for Authors
The current study employs theoretical analysis to construct a mathematical model based on a series of reasonable assumptions, with the aim of examining the likelihood of collisions between the floating offshore wind turbine (FOWT) carrier fleet and other vessels in the channel area during the towing process. The Channel Area Collision Analytical Model is developed using the Pedersen method, a commonly used and mature approach. However, the model significantly simplifies bend collision scenarios, and the Non-channel Area Collision Analytical Model employs the assumption of a relatively simple and uniform vessel density in the given maritime area. These simplifications may impact the accuracy of the subsequent analysis, and their adequacy is not thoroughly examined. As such, the innovation of the study is limited, and the analysis lacks depth.
However, the following recommendations are provided for the improvement of the manuscript:
1.Table 1 lists the research status of floating offshore wind turbines, which is not directly related to the topic of this article, namely, the reliability analysis of wind turbine and ship collisions. Therefore, this part should only be cited in the literature review section.
2.Section 2.2 introduces some assumptions for reliability analysis, including Alignment with Engineering Reality, Reduction of Data Requirement, and Specification of Usage Scenarios and Conditions. However, it is suggested that these basic contents be simplified and directly lead to subsequent assumptions.
3.The Results and Discussion section only presents collision frequencies for three months using the employed models, and whether the effectiveness of the model can be demonstrated with other indicators remains to be explored. Additionally, considering that the main focus of this paper is reliability analysis, the reliability of the Results and Discussion section seems to have been overlooked.
4.In the discussion section, there is too much descriptive text in the results section, and specific analytical data results are lacking. Therefore, it is recommended to provide more detailed and specific analytical data results to support the discussion
Comments on the Quality of English Languagecan be improved
Author Response
Please find the attached file, thanks.

Reviewer 4 Report
Comments and Suggestions for Authors
The manuscript proposed a novel theoretical model to examine the possibility of collisions between FOWT transport fleets and other vessels in the passage area during the towing process. The topic is interesting.
Proofreading should be done carefully to enhance the readability. For example, Line 50 is, in a way, difficult to understand, where ‘However’ is probably better replaced by ‘Fortunately’. Lines 235 to 237 seem to be bad examples of Chinglish.
Please carefully proofread the manuscript to avoid grammatical mistakes, typos, and any Chinglish usage.
There are several points that I believe should be addressed for this manuscript to be more beneficial for the readers:
Figure 5, Ship-fleet collision scenarios are not always just limited to two dimensions. Can the height of vessels and the influence of water depth be included in the model?
It is advised to provide an algorithm flow chart for the proposed model, as the reliability model proposed here involved too many parameters and assumptions.
Validation is absent in the manuscript, which should be improved in whole or in part. The author argued, ‘since the collision model for crossing navigation channels is further developed from the Pedersen model, its reliability has been widely recognized by the industry.’ However, these widely recognized results cannot lead to trust in the results here. A numerical simulation is recommended.
In a scientific journal, the conclusions in Lines 418 to 424 should be obtained from parameter sensitivity analysis rather than just qualitatively presented, which is not convincing at all. This applies to the suggestions provided in Lines 427 to 436.
The results are interesting. However, if the imposed implication can be further revealed or measures can be taken to counter the collisions, the results reported here would be more useful to the offshore wind project.
In lines 443-445, conclusions should be obtained by results discussions, and the conclusion should be supported by the results in a place. The readers were not expected to summarize the findings by themselves in a paper.
The fonts and their size in the figures or tables should be identical.
References should be in English only. Furthermore, the authors should check the bibliography to avoid any inconsistency with this journal.
Comments on the Quality of English Language
The English is poor. Moderate editing of English language required.
Author Response
Please find the attached file, thanks.

Reviewer 5 Report
Comments and Suggestions for Authors
The collision probablity of floating wind turbines during towing operation is an important topic. Thus, this article will make a contribution to the field if the results are valid. My major concerns are as follows,
1) The keywords should be improved and made more precise. mathematical model is too broad.
2) Fig.1, Fig.2, and Fig.3 are all taken from Reference [7] and not drawn by the authors. This should be highlighted in the figure capations with copyright permissions obtained.
3) 2.1 Case & site conditions should be prepared with a better structure. A description of the site condition can be grouped under case study.
4) The proposed probablity model for collision estimate is interesting. However, this model has not been verified by any real data. This is a weakness of the paper and should be addressed.
5) The discussion part should be structured in a better manner.
Comments on the Quality of English LanguageMinor English edits are needed. Refer to the discussion.
Author Response
Please find the attached file, thanks.

Round 2
Reviewer 3 Report
Comments and Suggestions for Authors
I have carefully considered your feedback and found you have made significant revisions to address the identified issues. I suggest to you expand the discussion section to include practical implications and possible areas of improvement for future engineers working. These additions aim to enhance the practical value of your study.
Comments on the Quality of English Languageno
Author Response
Thank you very much for your advice. The discussion section has been further extension based on your suggestions. The extension includes an analysis of the primary and secondary factors to be considered in the establishment of such collision models, as well as the limitations of such research. For details, please read lines 414 to 447.

Reviewer 5 Report
Comments and Suggestions for Authors
No further comments.
Author Response
Thank you for your acknowledgment! The revised manuscript can be found in the attachment.